# Causal Identification with Matrix Equations

**Sanghack Lee**
Graduate School of Data Science
Seoul National University
Seoul, South Korea
sanghack@snu.ac.kr

**Elias Bareinboim**
Department of Computer Science
Columbia University
New York, USA
eb@cs.columbia.edu

## Abstract

Causal effect identification is concerned with determining whether a causal effect is computable from a combination of qualitative assumptions about the underlying system (e.g., a causal graph) and distributions collected from this system. Many identification algorithms exclusively rely on graphical criteria made of a non-trivial combination of probability axioms, do-calculus, and refined c-factorization (e.g., Lee & Bareinboim, 2020). In a sequence of increasingly sophisticated results, it has been shown how *proxy* variables can be used to identify certain effects that would not be otherwise recoverable in challenging scenarios through solving matrix equations (e.g., Kuroki & Pearl, 2014; Miao et al., 2018). In this paper, we develop a new causal identification algorithm which utilizes both graphical criteria and matrix equations. Specifically, we first characterize the relationships between certain graphically-driven formulae and matrix multiplications. With such characterizations, we broaden the spectrum of proxy variable based identification conditions and further propose novel *intermediary* criteria based on the pseudoinverse of a matrix. Finally, we devise a causal effect identification algorithm, which accepts as input a collection of marginal, conditional, and interventional distributions, integrating enriched matrix-based criteria into a graphical identification approach.

## 1 Introduction

Cause and effect relations are one of the most common types of knowledge sought after throughout the empirical sciences. These relations are one of the main ingredients in the construction of stable explanations, and usually underpin robust and generalizable decision-making strategies [19, 26]. There is a growing literature that aims to systematically find causal relations by fusing observations, experiments, and substantive knowledge about the phenomenon under investigation [19, 3, 20]. Formally, the inferential target usually appears as the effect of a set of variables $do(\mathbf{X} = \mathbf{x})$ on another set of variables $\mathbf{Y}$, which is written as $P(\mathbf{y}|do(\mathbf{x}))$ or $P_{\mathbf{x}}(\mathbf{y})$. Assumptions about the underlying data-generating processes are commonly expressed as a causal graph $\mathcal{G}$ over endogenous variables $\mathbf{V}$. There are different lines of investigation that aims to establish the quantity $P_{\mathbf{x}}(\mathbf{y})$.

First, one line of investigation attempts to exploit the non-parametric constraints encoded in $\mathcal{G}$ to determine whether the quantity $P_{\mathbf{x}}(\mathbf{y})$ (i.e., query) is uniquely computable from the different available distributions. For instance, Pearl's celebrated method known as *do-calculus* provides a symbolic way of determining whether a query can be identified from $\mathcal{G}$ and observational data $P(\mathbf{V})$ [18]. A number of necessary and sufficient conditions were developed for systematically determining the identifiability status of the query from observational data [18, 29, 23, 9]. For example, the causal effect $P_x(y)$ in Fig. 1a is identified by a back-door criterion as $P_x(y) = \sum_u P(y|x, u)P(u)$. Further methods were developed to identify the query from a combination of observational and experimental distributions [2, 14, 12, 11]. The key ideas behind these methods are (i) to decompose the given causal query into standard *factors* (following Tian's *c-factorization*, [29]), leveraging the graphical

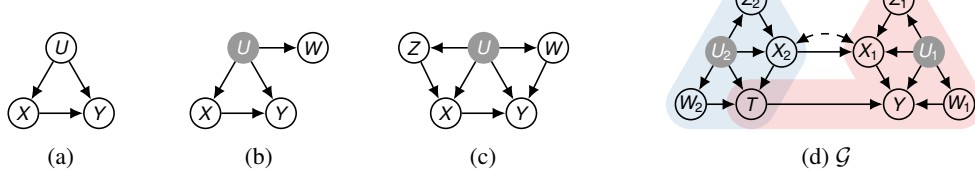

Figure 1: Causal graphs: (a) with a back-door condition; (b) with $W$ as a proxy for $U$; (c) with proxies $W$ and $Z$ for $U$. (d) A causal graph $\mathcal{G}$ where a causal effect $P_{\mathbf{x}}(y)$ can be computed by estimating each of $P_{x_2}(t)$ (light blue, left) and $P_{t,x_1}(y)$ (light red, right) using MGT criterion.

constraints, and (ii) to identify these factors individually, matching one of the available distributions. We call this the factorization-based approach.

Another line of investigation attempts to exploit assumptions about the relationship between unobserved confounders and the observable variables through the idea of proxy variables [5, 8, 10, 15, 32]. These methods rely on the cardinality and complexity of these relationships, which will possibly lead to the invertibility of certain matrices. The focus is usually on local conditions involving the treatment $X$, the outcome $Y$, and the proxies $W, Z$ for the set of unobserved confounders $U$; for example, see Fig. 1b. Note that since $U$ is unobserved, the effect of $X$ on $Y$, i.e., $P_x(y)$, is provably not identifiable by standard graphical methods discussed earlier. Still, if the distribution of the proxy given the unobserved confounder is available $P(W|U)$, in addition to $P(X, Y, W)$, the quantity $P_x(y)$ can be computed under the invertibility of $\mathbf{P}(W|U)$, a matrix representation of $P(W|U)$ where $\mathbf{P}(W|U)_{i,j} = P(w_i|u_j)$ (see Eq. (3) for detail). In case of Fig. 1c, the causal effect $P_x(y)$ is identified from $P(X, Y, W, Z)$ through the following.[1]

$$P_x(y) = \mathbf{P}(y|Z,x)\mathbf{P}(W|Z,x)^{-1}\mathbf{P}(W), \qquad (1)$$

where $\mathbf{P}(W|Z,x)$ has rank matching the size of domain of the unmeasured confounder $U$. Although typically not framed in terms of identification with multiple datasets, this setting corresponds to identifying $P_x(y)$ with, e.g., marginal distributions $\{P(X, Y, Z), P(W, X, Z)\}$ or conditional distributions $\{P(Y|Z,x), P(W|Z,x), P(W)\}$. We call this the proxy approach.[2]

Despite all the power and successes achieved by the factorization and proxy approaches, there exist still interesting cases not covered by any of them individually. To witness, consider the causal graph in Fig. 1d and first notice that the effect is *not* identified from each approach. On the other hand, if we combine both approaches, we are able to obtain the causal effect is identifiable and given by,

$$\begin{aligned}
P_{\mathbf{x}}(y) &= \sum_t P_{\mathbf{x}}(t,y) = \sum_t P_{t,x_1}(y)P_{x_2}(t) \\
&= \sum_t \left[\mathbf{P}(y|Z_1,x_1,t)\mathbf{P}(W_1|Z_1,x_1,t)^{-1}\mathbf{P}(W_1)\right]\left[\mathbf{P}(t|Z_2,x_2)\mathbf{P}(W_2|Z_2,x_2)^{-1}\mathbf{P}(W_2)\right],
\end{aligned}$$

where $P_{t,x_1}(y)$ and $P_{x_2}(t)$, the two factors of the query, are individually identified with the help of the proxy approach, i.e., Eq. (1).

Our goal in this paper is to explicate how this can be accomplished from first principles. More broadly, we will develop a flexible and general graphical identification approach that combines both the factorization and matrix equations (the underpinning idea of proxy-based methods). More specifically, our contributions are as follows:

1. We connect the graphical and matrical approaches by characterizing matrix equations of probability distributions driven by graphical constraints in a causal graph. This leads to a better understanding of the identification through solving systems of equations.

2. Building on this new characterization, we generalize proxy-based criteria and devise novel intermediary pseudoinverse criteria so as to identify a causal effect by utilizing the general inverse of a matrix and diverse collection of distributions.

---

[1]We focus on discrete variables. Detailed discussion on a continuous case can be found in [15]. We refer this identification condition *MGT criterion* where the acronym MGT comes from the surnames of the authors in [15].

[2]Recent work in this line of research use the term *proximal causal inference* (PCI) [25, 27, 33, 6]. [27] generalizes the results in [15] to allow observed confounders as well. [25] considered combining graphical and proximal approaches, similar to the motivational example.

3. We develop a general identification algorithm that amalgamates graphical and matrical approaches, returning an identification formula for a causal query given a causal diagram and a set of marginal, experimental, and conditional distributions. We show that this method subsumes current state of the art in the literature.

## 2 Preliminaries

We follow notational conventions from literature on causal inference. We denote a variable by an upper case letter $Y$, and its value is denoted by its corresponding lower case letter $y$ in the domain $\mathfrak{X}_Y$. A set of variables will be denoted by a bold capital letter $\mathbf{Y}$ with its value $\mathbf{y}$. We may use $\dot{\cup}$, instead of $\cup$, to emphasize the union of two *disjoint* sets. Given $\mathbf{Z} \subseteq \mathbf{W}$, $\mathbf{w} \backslash \mathbf{Z}$ denotes the value of $\mathbf{W} \backslash \mathbf{Z}$ consistent with $\mathbf{w}$. Let $\mathbf{a}/\mathbf{B} = (\mathbf{A} \cap \mathbf{B}, \mathbf{a} \backslash \mathbf{B})$ which retains $\mathbf{B}$ as a set of variables and values of $\mathbf{a}$ excluding $\mathbf{B}$. Without loss of generality, we refer $P_{\mathbf{Z}}(\mathbf{V}'|\mathbf{W})$ a distribution. We may employ *conditional* to emphasize $\mathbf{W} \supseteq \emptyset$, *experimental* or *interventional* $\mathbf{Z} \supseteq \emptyset$ compared to *observational* $\mathbf{Z} = \emptyset$, and *marginal* if $\mathbf{Z} \cup \mathbf{W} \cup \mathbf{V}' \subsetneq \mathbf{V}$.

We employ structural causal models (SCMs) [19, Ch. 7] as the semantical framework to represent a domain of interest. An SCM $\mathcal{M}$ is a quadruple $\langle \mathbf{U}, \mathbf{V}, P(\mathbf{U}), \mathbf{F} \rangle$. A set of exogenous variables $\mathbf{U}$, which follows $P(\mathbf{U})$, is determined by factors outside the model. $\mathbf{V}$ is a set of endogenous variables whose values are determined by functions $\mathbf{F} = \{f_i\}_{V_i \in \mathbf{V}}$ such that $V_i \leftarrow f_i(\mathbf{pa}_i, \mathbf{u}_i)$ where $\mathbf{PA}_i \subseteq \mathbf{V} \backslash \{V_i\}$ and $\mathbf{U}_i \subseteq \mathbf{U}$. Further, $do(\mathbf{x})$ represents the operation of holding a set $\mathbf{X}$ to a constant $\mathbf{x}$ regardless of their original mechanisms. Such intervention induces a submodel $\mathcal{M}_{\mathbf{x}}$, which is $\mathcal{M}$ with $f_X$ replaced to $x$ for $X \in \mathbf{X}$. The distribution over $\mathbf{V}$ induced by the submodel is denoted by $P_{\mathbf{x}}(\mathbf{V})$. We may employ letter $Q$ to denote an interventional distribution, e.g., $Q = P_{\mathbf{r}}$.

Each SCM (model, for short) induces a causal diagram (or causal graph) $\mathcal{G} = \langle \mathbf{V}, \mathbf{E} \rangle$, where each type of edge represents a different causal relationship among the variables: (i) $X \rightarrow Y$ if $X$ is used as an argument of $f_Y$; and (ii) $X \leftrightarrow Y$ if $\mathbf{U}_X$ and $\mathbf{U}_Y$ are correlated. Given a causal diagram $\mathcal{G}$, familial relationships among its vertices are denoted by $\mathrm{pa}$ and $\mathrm{an}$ for parents and ancestors, respectively. Further, $\mathrm{An}$ is a set of ancestors including its argument as well. We denote by $\mathcal{G}_{\overline{\mathbf{X}}\underline{\mathbf{Z}}}$ an edge subgraph of $\mathcal{G}$ which removes edges incoming to $\mathbf{X}$ and outgoing from $\mathbf{Z}$. Causal relationships among other variables are captured in $\mathcal{G} \backslash \mathbf{X}$, which is the subgraph of $\mathcal{G}$ over $\mathbf{V} \backslash \mathbf{X}$. A vertex induced subgraph is denoted by $\mathcal{G}[\mathbf{V}']$ where $\mathbf{V}' \subseteq \mathbf{V}$. Causal effect identification relies heavily on standard graphical constraints imposed by a causal diagram such as d-separation (reading off conditional independence from the graph, [30, 7]) and do-calculus (equivalence among interventional probabilities) [18], which we present in the Appendix for completeness.

The *latent projection* (or projection, for short) of a causal diagram is a causal diagram retaining the causal relationships among a subset of variables. We denote by $\mathcal{G}\langle \mathbf{V}' \rangle$ the latent projection of $\mathcal{G}$ onto $\mathbf{V}' \subseteq \mathbf{V}$, the causal graph over $\mathbf{V}'$ [31]. Conditional independence (CI) statements and do-calculus [18] on a projection are valid in $\mathcal{G}$, vice versa. We formally define a latent projection in the supplementary material. The omitted proofs and derivations are also provided in [13].

## 3 Characterization of Matrix Equations of Graphical Constraints

In this section, we present characterizations of graphical constraints in a given causal diagram $\mathcal{G}$, leading to equations expressed as the multiplication of matrices. The characterizations will further advance our understanding on the constraints imposed over the distributions generated by the underlying system compared to simple equivalence relationships such as conditional independence and do-calculus. To begin with, we denote by $\mathbf{P}$ the matrix representation of a distribution $P$ where free outcome variables are rows and free condition or intervention variables correspond to columns. For instance, $\mathbf{P}_{a,B}(C, d)$ is a $|\mathfrak{X}_C| \times |\mathfrak{X}_B|$ matrix and $\mathbf{P}_r(A, b|C, d)$ is a $|\mathfrak{X}_A| \times |\mathfrak{X}_C|$ matrix. Further, we may use $'$ and $''$ to represent two disjoint subsets such that $\mathbf{B} = \mathbf{B}' \dot{\cup} \mathbf{B}''$.

**Chain Rule with Conditional Independence** The definitions of conditional and marginal distributions naturally lead to a sum of a product of probabilities. Let $Q = P_{\mathbf{r}}$ be an arbitrary interventional distribution. Let $\mathbf{A}$, $\mathbf{B}$, $\mathbf{C}$, and $\mathbf{R}$ be disjoint. A marginal probability over chain-rule-induced multi-

plication is expressed as

$$Q(\mathbf{a}, \mathbf{b}'|\mathbf{c}) = \sum_{\mathbf{b}''} Q(\mathbf{a}|\mathbf{b}, \mathbf{c})Q(\mathbf{b}|\mathbf{c}) = \mathbf{Q}(\mathbf{a}|\mathbf{b}', \mathbf{B}'', \mathbf{c})\mathbf{Q}(\mathbf{B}'', \mathbf{b}'|\mathbf{c}).$$

Considering conditional independence, we can further enrich such a characterization.

**Lemma 1.** *Given a causal diagram $\mathcal{G}$, let $Q = P_{\mathbf{r}}$ for some $\mathbf{r} \in \mathfrak{X}_{\mathbf{R}}$ where $\mathbf{R} \subsetneq \mathbf{V}$. Let $\mathbf{A}, \mathbf{B}, \mathbf{C}, \mathbf{D}, \mathbf{E}$ be disjoint subsets of $\mathbf{V} \backslash \mathbf{R}$. If $(\mathbf{D} \perp\!\!\!\perp \mathbf{A} \mid \mathbf{B}, \mathbf{C}, \mathbf{E})$ and $(\mathbf{E} \perp\!\!\!\perp \mathbf{B} \mid \mathbf{C}, \mathbf{D})$ in $\mathcal{G} \backslash \mathbf{R}$, then, $\mathbf{Q}(\mathbf{A}, \mathbf{b}'|\mathbf{c}, \mathbf{D}, \mathbf{e}) = \mathbf{Q}(\mathbf{A}|\mathbf{b}', \mathbf{B}'', \mathbf{c}, \mathbf{e})\mathbf{Q}(\mathbf{B}'', \mathbf{b}'|\mathbf{c}, \mathbf{D}).$*

The lemma emphasizes the condition under which the result of multiplication is a matrix not just a row or column. A special case of the lemma is appeared in MGT criterion where the inverse of a matrix multiplication $\mathbf{P}(W|Z, x) = \mathbf{P}(W|U, x)\mathbf{P}(U|Z, x)$ is utilized so as to cancel out an unknown distribution $\mathbf{P}(W|U) = \mathbf{P}(W|U, x)$.

**Adjustment Criterion**  Given a graph $\mathcal{G}$ and a causal effect of interest $P_{\mathbf{x}}(\mathbf{y})$, the adjustment criterion [24] seeks a set of covariates $\mathbf{Z} \subseteq \mathbf{V} \backslash \mathbf{X} \backslash \mathbf{Y}$, called an *adjustment set* for a causal effect $P_{\mathbf{x}}(\mathbf{y})$, which grants the following expression, $P_{\mathbf{x}}(\mathbf{y}) = \sum_{\mathbf{z}} P(\mathbf{y}|\mathbf{x}, \mathbf{z})P(\mathbf{z})$. Adjustment criterion generalizes back-door criterion [19]. Its matricized expression with employing $Q = P_{\mathbf{r}}$ is

$$Q_{\mathbf{x}}(\mathbf{y}) = \sum_{\mathbf{z}} Q(\mathbf{y}|\mathbf{x}, \mathbf{z})Q(\mathbf{z}) = \mathbf{Q}(\mathbf{y}|\mathbf{x}, \mathbf{Z})\mathbf{Q}(\mathbf{Z}).$$

This simple expression plays a central role in the identification with proxy variables.

In many settings, the left hand side (LHS) is the query of interest and two terms in the RHS are usually available or to be inferred using other available quantities. However, substituting value $\mathbf{y}$ with $\mathbf{Y}$, we can further yield (under invertibility assumption) $\mathbf{Q}(\mathbf{Z}) = \mathbf{Q}(\mathbf{Y}|\mathbf{x}, \mathbf{Z})^{-1}\mathbf{Q}_{\mathbf{x}}(\mathbf{Y})$, which restores the covariate distribution of interest given a causal effect and conditional distribution.

**C-Factorization**  C-factorization [28] decomposes a causal effect $P_{\mathbf{x}}(\mathbf{y})$ into the sum-product of c-factors (simply, factors) with respect to the given causal diagram $\mathcal{G}$. Without loss of generality, let $\mathbf{X}$ be minimal such that no $\mathbf{X}' \subsetneq \mathbf{X}$ satisfies $P_{\mathbf{x}'}(\mathbf{y}) \neq P_{\mathbf{x}}(\mathbf{y})$ (i.e., overriding $\mathbf{X}$ by $\mathrm{an}^{\mathcal{G}_{\overline{\mathbf{x}}}}(\mathbf{Y}) \cap \mathbf{X}$). For any projection $\mathcal{H}$ of $\mathcal{G}$ that preserves $\mathbf{X} \cup \mathbf{Y}$, the following holds:

$$P_{\mathbf{x}}(\mathbf{y}) = \sum_{\mathbf{y}^+ \backslash \mathbf{y}} P_{\mathbf{x}}(\mathbf{y}^+) = \sum_{\mathbf{y}^+ \backslash \mathbf{y}} \prod_{\mathbf{Y}_i \in \mathcal{C}(\mathcal{H}[\mathbf{Y}^+])} P_{\mathrm{pa}^{\mathcal{H}}(\mathbf{y}_i) \backslash \mathbf{y}_i}(\mathbf{y}_i), \tag{2}$$

where $\mathbf{Y}^+ = \mathrm{An}^{\mathcal{H}_{\overline{\mathbf{x}}}}(\mathbf{Y})$ and $\mathcal{C}(\cdot)$ is the c-component decomposition (partitioning the variables in the graph based on their connectivity through bidirected edges). Given a query $P_{\mathbf{x}}(\mathbf{y})$, we let c-factors $\mathbb{F}_{\mathcal{H}} = \{\langle \mathbf{X}_i, \mathbf{Y}_i \rangle\}_i$ where $\{\mathbf{Y}_i\}_i$ form the c-component decomposition of $\mathcal{H}[\mathbf{Y}^+]$ and $\mathbf{X}_i = \mathrm{pa}^{\mathcal{H}}(\mathbf{Y}_i) \backslash \mathbf{Y}_i$. The general identification method under partial-observability [12] can be summarily described as finding $\mathcal{H}$ such that each factor is identified by one of the available distributions.

Let us denote $\mathbf{X}_{ij} = \mathbf{X}_i \cup \mathbf{X}_j$. We focus here on the sum-product of a pair of factors at some latent projection $\mathcal{H}$, which satisfies $P_{\mathbf{x}_i}(\mathbf{y}_i)P_{\mathbf{x}_j}(\mathbf{y}_j) = P_{\mathbf{x}_{ij} \backslash \mathbf{y}_{ij}}(\mathbf{y}_{ij})$ where the values are consistent [12]. Now consider the summation over $\mathbf{Z} \subseteq \mathbf{X}_i \cap \mathbf{Y}_j$. Then,

$$\sum_{\mathbf{z}} P_{\mathbf{x}_i \backslash \mathbf{z}, \mathbf{z}}(\mathbf{y}_i)P_{\mathbf{x}_j}(\mathbf{y}_j \backslash \mathbf{z}, \mathbf{z}) = P_{\mathbf{x}_{ij} \backslash \mathbf{y}_{ij}}(\mathbf{y}_{ij} \backslash \mathbf{Z}).$$

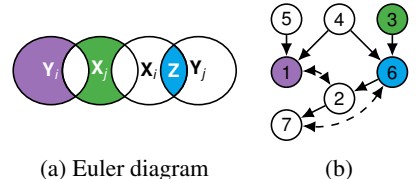

(a) Euler diagram          (b)

Figure 2: (a) set relationships, (b) a causal graph where the numbers correspond to the regions of (a) from the left to right.

To properly represent this as a matrix multiplication, we should decide which variables will become rows and columns. Two matrices to be multiplied should share only $\mathbf{Z}$, and other common variables, appearing in the two terms, or non-matching variables, appearing as intervention in left term and outcome in right term, need to be set to constants. To help understand, we illustrate in Fig. 2a the set relationships among $\mathbf{X}_i, \mathbf{Y}_i, \mathbf{X}_j$, and $\mathbf{Y}_j$ in the case of $\mathbf{Z} = \mathbf{X}_i \cap \mathbf{Y}_j$.

Therefore, $\mathbf{Y}_i \cap \mathbf{X}_j$ (the shared variables other than $\mathbf{Z}$) and $(\mathbf{X}_i \cup \mathbf{Y}_j) \backslash \mathbf{Z}$ (the non-matching variables) are fixed. These fixed variables are not colored in Fig. 2a. As a result, we can obtain a submatrix of $\mathbf{P}_{\mathbf{X}_{ij} \backslash \mathbf{Y}_{ij}}(\mathbf{Y}_{ij} \backslash \mathbf{Z})$ as the multiplication of the submatrices of $\mathbf{P}_{\mathbf{X}_i}(\mathbf{Y}_i)$ and $\mathbf{P}_{\mathbf{X}_j}(\mathbf{Y}_j)$ obtained via fixing variables.

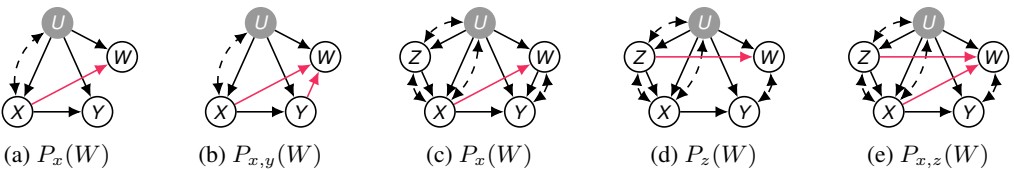

| (a) $P_x(W)$ | (b) $P_{x,y}(W)$ | (c) $P_x(W)$ | (d) $P_z(W)$ | (e) $P_{x,z}(W)$ |

Figure 3: Causal diagrams admitting single- and double-proxy settings with surrogate experiments.

**Lemma 2** (Matrix Equation of C-Factorization with Two Factors). *Given a causal diagram $\mathcal{G}$ and an experimental distribution $Q = P_{\mathbf{r}}$ where a causal effect is c-factorized as $Q_{\mathbf{x}}(\mathbf{y}) = \sum_{\mathbf{z}} Q_{\mathbf{x}_i}(\mathbf{y}_i) Q_{\mathbf{x}_j}(\mathbf{y}_j)$ in a projection $\mathcal{H}$ of $\mathcal{G} \backslash \mathbf{R}$, the effect can be represented as a matrix multiplication, if $\mathbf{Z} \subseteq \mathbf{X}_i \cap \mathbf{Y}_j$. Further, the corresponding matrix equation is*

$$\mathbf{Q}_{(\mathbf{x}_{ij} \backslash \mathbf{y}_{ij})/(\mathbf{X}_j \backslash \mathbf{X}_i \backslash \mathbf{Y}_i)}((\mathbf{y}_{ij} \backslash \mathbf{Z})/(\mathbf{Y}_i \backslash \mathbf{X}_j)) = \mathbf{Q}_{\mathbf{x}_i/\mathbf{Z}}(\mathbf{y}_i/(\mathbf{Y}_i \backslash \mathbf{X}_j)) \mathbf{Q}_{\mathbf{x}_j/(\mathbf{X}_j \backslash \mathbf{X}_i \backslash \mathbf{Y}_i)}(\mathbf{y}_j/\mathbf{Z}).$$

We illustrate a causal graph in Fig. 2b where each variable matches to each region in the Euler diagram (Fig. 2a). With $\mathbf{Z} = \{6\}$ and for an arbitrary instantiation of variables $V_2, V_4, V_5,$ and $V_7$, $\mathbf{P}_{V_3, v_4, v_5}(V_1, v_2, v_7) = \mathbf{P}_{v_4, v_5, V_6}(V_1, v_2) \mathbf{P}_{v_2, V_3, v_4}(V_6, v_7)$.

In this section, we connected different graphical constraints—from chain-rule with conditional independence to c-factorization—induced identification formulae to matrix equations. Results presented in this section are by no means complete yet cover, to the best of our knowledge, every sum-product expression appeared in both graphical and matrical identification approaches. Nevertheless, this suite of characterizations will provide a fundamental understanding of the mathematical structures involved in the identification methods with matrical expressions.

## 4 Generalized Proxy-based Criteria

Equipped with the characterization from the previous section, we revisit single- and double-proxy settings [5, 8, 10, 15] more formally, which identify a causal effect through the combination of chain rule, adjustment criterion (c-factorization), and inverses of matrices given an observational distribution and additional external study involving an unobserved confounder. We investigate its extension capable of utilizing other types of available distributions. Results presented in this section is crucial in adopting matrical approaches into a factorization-based identification algorithm.

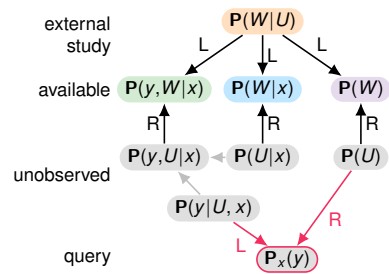

Figure 4: Schematic for a single proxy setting. Gray and red lines are for elementwise matrix multiplication and adjustment criterion, respectively.

**Single-Proxy Setting** We illustrate in Fig. 4 the available distributions and unknown distributions, considered in Fig. 1b, that can lead to $P_x(y)$. In the figure, a matrix multiplication $A = B \cdot C$ is represented as $B \xrightarrow{\text{L}} A \xleftarrow{\text{R}} C$ with positions annotated. First, note that $(X, Y \perp\!\!\!\perp W \mid U)$ in a causal graph $\mathcal{G}$ is central, which grants $P(W|U, x, y) = P(W|U, x) = P(W|U)$. If $\mathbf{P}(W|U)$ is invertible,[3] the three distributions $\mathbf{P}(y, U|x)$, $\mathbf{P}(U|x)$, and $\mathbf{P}(U)$ are obtained where $\mathbf{P}(y|U, x)$ is computed by chain rule (Lemma 1). Then, $P_x(y)$ is, with $\oslash$ denoting elementwise division,

$$\overbrace{\left( \underbrace{(\mathbf{P}(W|U)^{-1}\mathbf{P}(y, W|x))}_{\mathbf{P}(y,U|x)} \oslash \underbrace{(\mathbf{P}(W|U)^{-1}\mathbf{P}(W|x))}_{\mathbf{P}(U|x)} \right)^{\top}}^{\mathbf{P}(y|U,x)} \underbrace{\mathbf{P}(W|U)^{-1}\mathbf{P}(W)}_{\mathbf{P}(U)}. \tag{3}$$

However, this scheme would not work in more restrictive conditions such as Fig. 3a or 3b. This challenging scenario can be handled if a different external study is available altogether with a surrogate

---

[3]In case of $W$ has more categorical values than $U$, one can coarsen $W$ to $W'$ ensuring that $\mathbf{P}(W'|U)$ is full rank. In other words, $\mathbf{P}(W|U)$ has full column rank.

experiment. For example, when only $Y$ is independent to $W$ given $U$ and $X$, the availability of an external study $P(W|U,x)$, instead of $P(W|U)$, and a surrogate experiment $P_x(W)$, in addition to the observational distribution $P(Y,W,X)$, suffices to identify the causal effect (see [13] for the derivations for Fig. 3a, 3b). These examples suggest that we can generalize the single proxy-based criterion to take advantage of a diverse collection of external studies and surrogate experiments.

**Theorem 1.** *Given $\mathcal{G}$, let $\mathbf{X}$, $\mathbf{Y}$, $\mathbf{W}$, $\mathbf{U}$, and $\mathbf{R}$ be disjoint subsets of $\mathbf{V}$, $Q = P_{\mathbf{r}}$ for some $\mathbf{r} \in \mathfrak{X}_{\mathbf{R}}$, and $\mathcal{H} = \mathcal{G}\backslash\mathbf{R}$. A causal effect $P_{\mathbf{r},\mathbf{x}}(\mathbf{y}) = Q_{\mathbf{x}}(\mathbf{y})$ is identifiable if (1) $\mathbf{U}$ is an adjustment admissible set for $Q_{\mathbf{x}}(\mathbf{y})$ in $\mathcal{H}$; (2a) $(\mathbf{Y} \perp\!\!\!\perp \mathbf{W} \mid \mathbf{U},\mathbf{X})_{\mathcal{H}}$ and $Q(\mathbf{W}|\mathbf{U},\mathbf{x})$, $Q(\mathbf{y},\mathbf{W}|\mathbf{x})$ and $Q(\mathbf{W}|\mathbf{x})$ are available where $\mathbf{Q}(\mathbf{W}|\mathbf{U},\mathbf{x})$ has full column rank; or (2b) $(\mathbf{Y} \not\perp\!\!\!\perp \mathbf{W} \mid \mathbf{U},\mathbf{X})_{\mathcal{H}}$ and $Q(\mathbf{W}|\mathbf{U},\mathbf{x},\mathbf{Y})$ and $Q(\mathbf{Y},\mathbf{W}|\mathbf{x})$ are available where every $\mathbf{Q}(\mathbf{W}|\mathbf{U},\mathbf{x},\mathbf{y}')$ has full column rank for $\mathbf{y}' \in \mathfrak{X}_{\mathbf{Y}}$; (3) $Q_{\mathbf{z}'}(\mathbf{W})$ is available for $\mathbf{Z} \subseteq \mathbf{X} \cup \mathbf{Y}$ and $\mathbf{Z}' = (\mathbf{X} \cup \mathbf{Y})\backslash\mathbf{Z}$ with $\mathbf{z}'$ consistent with $\mathbf{x} \cup \mathbf{y}$ such that $(\mathbf{Z} \perp\!\!\!\perp \mathbf{W} \mid \mathbf{U},\mathbf{Z}')$ in $\mathcal{H}$; and $\mathbf{U}$ is an adjustment admissible set for $Q_{\mathbf{z}'}(\mathbf{W})$ in $\mathcal{H}$.*

The theorem presents a sound condition to elicit the causal effect through combining various distributions especially when the given situation is more restrictive.

**Double-Proxy Setting** We now generalize MGT criterion to utilize distributions other than the originally considered observational study. MGT criterion for a double-proxy setting relies on the following conditions to identify $P_x(y)$ with $P(X,Y,Z,W)$:

(C1) $U$ is an adjustment set for $P_x(y)$ in $\mathcal{G}$;

(C2) $Y \perp\!\!\!\perp Z \mid U, X$ in $\mathcal{G}$;

(C3) $Z, X \perp\!\!\!\perp W \mid U$ in $\mathcal{G}$; and

(C4) $\mathbf{P}(W|Z,x)$ has rank $|\mathfrak{X}_U|$.

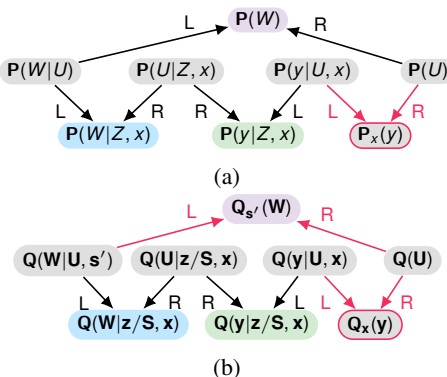

(a)

(b)

Figure 5: Schematics of (a) MGT criterion and (b) its generalization (simplified)

Under these conditions, algebraic relationships between a causal effect $P_x(y)$ and other distributions can be illustrated as in Fig. 5a where the distributions form a closed loop alternating between (i) distributions with an unmeasured confounder and (ii) given distributions and a query. Among the four multiplications, $P_x(y)$ corresponds to an adjustment criterion (C1) so as to $P_x(y) = \mathbf{P}(y|U,x)\mathbf{P}(U)$. Others are due to the chain-rule combined with CI, e.g., (C2) $\mathbf{P}(y|U,x) = \mathbf{P}(y|U,x,z)$ and (C3) $\mathbf{P}(W|U) = \mathbf{P}(W|U,x,z)$. With (C4), which implies that both $\mathbf{P}(W|U)$ and $\mathbf{P}(U|Z,x)$ are invertible under coarsening of $W$ and $Z$ if necessary [15, 1], the causal effect $P_x(y)$ can be expressed as Eq. (1) by subsequently rewriting $\mathbf{P}(U|Z,x)$ and $\mathbf{P}(y|U,x)$, and contracting $\mathbf{P}(W) = \mathbf{P}(W|U)\mathbf{P}(U)$ (see [13] Appendix C for an illustration).

To examine the possible extension of the setting, we relax (C3), where the CI grants the use of matrix multiplication leading to a chain-rule $\mathbf{P}(W) = \mathbf{P}(W|U)\mathbf{P}(U)$ and $\mathbf{P}(W|Z,x) = \mathbf{P}(W|U)\mathbf{P}(U|Z,x)$. One can consider three relaxed versions of (C3), namely, $(Z \perp\!\!\!\perp W \mid U, X)$, $(X \perp\!\!\!\perp W \mid U, Z)$, and, ultimately, dropping (C3) illustrated respectively in Fig. 3c, 3d and 3e. For concreteness, consider the first relaxation $(Z \perp\!\!\!\perp W \mid U, X)$. The assumption grants $\mathbf{P}(W|U,Z,x)=\mathbf{P}(W|U,x)$. Given that a surrogate experiment $P_x(W)$ is accessible, and it can be decomposed as $\mathbf{P}_x(W) = \mathbf{P}(W|U,x)\mathbf{P}(U)$ where $U$ is also an adjustment admissible set for $P_x(w)$, then, $P_x(y)$ is identified. Other two relaxations turned out to be much more challenging, yet the causal effect can be identified by exploiting different surrogate experiments or additional external study (see [13] Appendix C for detail). Motivated by these examples, we present a theorem that extends MGT criterion with varying degrees of the assumption and data collection.

**Theorem 2** (Generalized MGT Criterion). *Given a causal graph $\mathcal{G}$, let $\mathbf{X}, \mathbf{Y}, \mathbf{Z}, \mathbf{W}, \mathbf{U}, \mathbf{R} \subset \mathbf{V}$ be disjoint sets of variables where $\mathbf{R}$ can be empty. Let $Q = P_{\mathbf{r}}$ for some $\mathbf{r} \in \mathfrak{X}_{\mathbf{R}}$ and $\mathcal{H} = \mathcal{G}\backslash\mathbf{R}$. Let $\mathbf{S} \subseteq \mathbf{X}\cup\mathbf{Z}$. A causal effect $Q_{\mathbf{x}}(\mathbf{y}) = P_{\mathbf{x},\mathbf{r}}(\mathbf{y})$ is identifiable in $\mathcal{G}$ if, for some $\mathbf{z}$, (1) $\mathbf{U}$ is an adjustment set for $Q_{\mathbf{x}}(\mathbf{y})$ in $\mathcal{H}$; (2) $(\mathbf{Y} \perp\!\!\!\perp \mathbf{Z} \mid \mathbf{U},\mathbf{X})_{\mathcal{H}}$; (3) $(\mathbf{W} \perp\!\!\!\perp \mathbf{S}' \mid \mathbf{U},\mathbf{S})_{\mathcal{H}}$ where $\mathbf{S}' = (\mathbf{X}\cup\mathbf{Z})\backslash\mathbf{S}$; (4) $\mathbf{U}$ is an adjustment set for $Q_{\mathbf{s}'}(\mathbf{W})$ in $\mathcal{H}$; (5) $\mathbf{Q}(\mathbf{W}|\mathbf{U},\mathbf{s}')$ is invertible; (6) $\mathbf{Q}(\mathbf{U}|\mathbf{z}/(\mathbf{S}\cup\mathbf{Z}'),\mathbf{x})$ is invertible for some $\mathbf{Z}' \subseteq \mathbf{Z}\backslash\mathbf{S}$ and $Q(\mathbf{y}|\mathbf{z}/(\mathbf{S}\cup\mathbf{Z}'),\mathbf{x})$, $Q(\mathbf{W}|\mathbf{z}/(\mathbf{S}\cup\mathbf{Z}'),\mathbf{x})$, and $Q_{\mathbf{s}'}(\mathbf{W})$ are available. Additionally, $Q(\mathbf{W}|\mathbf{U},\mathbf{s}'/\mathbf{Z}')$ is available if $\mathbf{Z}' \neq \emptyset$.*

The theorem broadens the applicability of the MGT criterion as an identification template to a range of collection of distributions. Although the conditions involved in generalizing MGT criterion non-trivial, we can similarly draw a (simplified) big picture of the generalized criterion as in Fig. 5b. The key idea is relaxing the C3 condition for a set of variables while considering sets of variables (explicitly) and interventional distributions.

We have established generalized identification criteria exploiting proxy variables.[4] The purpose of both criteria aligns well with the philosophy of general identification [14, 12] so that they can be smoothly integrated into graphical approaches.

## 5 Pseudoinverse and Intermediary Criteria

Now, we present a novel identification condition for a challenging setting that neither previous matrical nor graphical approaches could handle. In the setting, the probability of interest is expressed as the multiplication of *three* matrices and is identifiable through employing the *pseudoinverse* of a matrix, *without* an invertibility assumption. Let $\mathbf{P}(\cdot)^{\dagger}$ denote the pseudoinverse of $\mathbf{P}(\cdot)$.[5]

**Lemma 3** (Base Intermediary Criterion)**.** *Let* $\{P_1, P_2, P_3, P_4\}$ *be distributions. Let* $\{\mathbf{P}_i\}_{i=1}^{4}$ *be their matrix representations. If submatrices* $\{\mathbf{P}'_i\}_{i=1}^{4}$ *of* $\{\mathbf{P}_i\}_{i=1}^{4}$ *satisfy* $\mathbf{P}'_1 = \mathbf{P}'_2 \mathbf{P}'_3 \mathbf{P}'_4$ *and* $\mathbf{P}'_2 \mathbf{P}'_3$, $\mathbf{P}'_3 \mathbf{P}'_4$, *and* $\mathbf{P}'_3$ *are given, then,* $\mathbf{P}'_1 = (\mathbf{P}'_2 \mathbf{P}'_3) \mathbf{P}'_3{}^{\dagger} (\mathbf{P}'_3 \mathbf{P}'_4)$.

*Proof.* By the given condition, the associativity of matrix multiplications, and the property of pseudoinverse $\mathbf{P}\mathbf{P}^{\dagger}\mathbf{P} = \mathbf{P}$, $\mathbf{P}'_1 = \mathbf{P}'_2 \mathbf{P}'_3 \mathbf{P}'_4 = \mathbf{P}'_2 (\mathbf{P}'_3 \mathbf{P}'_3{}^{\dagger} \mathbf{P}'_3) \mathbf{P}'_4 = (\mathbf{P}'_2 \mathbf{P}'_3) \mathbf{P}'_3{}^{\dagger} (\mathbf{P}'_3 \mathbf{P}'_4)$. □

Although the lemma itself is rather general, we concretely characterize distributions satisfying Lemma 3 with respect to chain-rule (Sec. 5.1) and c-factorization (Sec. 5.2).

### 5.1 Chain-Rule-based Intermediary Criterion

We start by characterizing an intermediary criterion with a chain-rule using a simple illustrative example. Let $Q$ be an arbitrary interventional distribution. One way to decompose $Q(a, b, c|d)$ into three probabilities is $Q(a|b, c, d)Q(b|c, d)Q(c|d)$. Let a probability of interest be its marginal $Q(a|d) = \sum_{b,c} Q(a, b, c|d)$ where the following distributions are available: $Q(B|C, d)$, $Q(A|C, d)$, and $Q(B|d)$. If the first term $Q(a|b, c, d)$ is equal to $Q(a|b, d)$, the term can be multiplied by $Q(b|d) = \sum_c Q(b|c, d)Q(c|d)$. Hence, the matricized expression becomes,

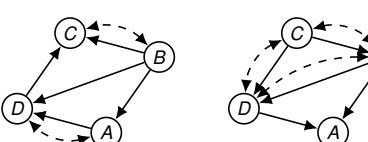

Figure 6: Causal diagrams where chain-rule intermediary criterion is applicable to identify $P(a|d)$ given $\mathbf{P}(a|C, d)$, $\mathbf{P}(B|C, d)$, and $\mathbf{P}(B|d)$.

$$\mathbf{Q}(A|d) = \overbrace{\mathbf{Q}(A|B, d) \underbrace{\mathbf{Q}(B|C, d)}_{\mathbf{Q}(B|d)} \mathbf{Q}(C|d)}^{\mathbf{Q}(A|C,d)} = \mathbf{Q}(A|C, d)\mathbf{Q}(B|C, d)^{\dagger}\mathbf{Q}(B|d).$$

for any $d \in \mathfrak{X}_D$. Two illustrative examples where this expression is applicable (i.e., $(C \perp\!\!\!\perp A \mid B, D)$ in $\mathcal{G}$) are shown in Fig. 6. Now, we propose a chain-rule intermediary criterion.

**Lemma 4** (Chain-Rule Intermediary Criterion)**.** *Given a causal diagram* $\mathcal{G}$, *let* $\mathbf{A}$, $\mathbf{B}$, $\mathbf{C}$, $\mathbf{D}$, *and* $\mathbf{R}$ *be disjoint subsets of* $\mathbf{V}$ *with* $\mathbf{D}$ *and* $\mathbf{R}$ *can be empty. Let* $\mathbf{B} = \mathbf{B}' \dot{\cup} \mathbf{B}''$ *and* $\mathbf{C} = \mathbf{C}' \dot{\cup} \mathbf{C}''$ *where* $\mathbf{B}'$ *and* $\mathbf{C}'$ *are not empty. Given an interventional distribution* $Q = P_{\mathbf{r}}$, *if* $(\mathbf{C}' \perp\!\!\!\perp \mathbf{A} \mid \mathbf{B}, \mathbf{C}''\mathbf{D})$ *in* $\mathcal{G} \setminus \mathbf{R}$ *and* $Q(\mathbf{a}, \mathbf{b}'', \mathbf{c}''|\mathbf{d}) = \sum_{\mathbf{b}', \mathbf{c}'} Q(\mathbf{a}|\mathbf{b}, \mathbf{c}, \mathbf{d})Q(\mathbf{b}|\mathbf{c}, \mathbf{d})Q(\mathbf{c}|\mathbf{d})$, *then,*

$$\mathbf{Q}(\mathbf{A}, \mathbf{b}'', \mathbf{c}''|\mathbf{d}) = \mathbf{Q}(\mathbf{A}, \mathbf{b}''|\mathbf{C}', \mathbf{c}'', \mathbf{d}) \cdot \mathbf{Q}(\mathbf{B}', \mathbf{b}''|\mathbf{C}', \mathbf{c}'', \mathbf{d})^{\dagger} \cdot \mathbf{Q}(\mathbf{B}', \mathbf{b}'', \mathbf{c}''|\mathbf{d}).$$

---

[4]As mentioned earlier, [27] extends the MGT criterion by allowing observed confounders $C$. Roughly, this can be understood as replacing $U$ in Fig. 5a with $U$ and $C$ where $C$ is not necessarily marginalized out so that the resulting distribution may contain $C$.

[5]In case of continuous variables, $\mathbf{P}(B|A)$ can be understood as a linear operator on Hilbert spaces. See [16] for more detailed discussion on the existence and uniqueness of pseudoinverse for a bounded linear operator.

Additional CI allows the criterion to return a matrix, which can then be nested into other equation.

**Corollary 1.** *Given Lemma [4], if* $(\mathbf{D} \perp\!\!\!\perp \mathbf{A}, \mathbf{B}'' \mid \mathbf{C})$ *and* $(\mathbf{D} \perp\!\!\!\perp \mathbf{B} \mid \mathbf{C})$ *in* $\mathcal{G} \backslash \mathbf{R}$*, then*

$$\mathbf{Q}(\mathbf{A}, \mathbf{b}'', \mathbf{c}'' | \mathbf{D}) = \mathbf{Q}(\mathbf{A}, \mathbf{b}'' | \mathbf{C}', \mathbf{c}'') \cdot \mathbf{Q}(\mathbf{B}', \mathbf{b}'' | \mathbf{C}', \mathbf{c}'')^{\dagger} \cdot \mathbf{Q}(\mathbf{B}', \mathbf{b}'', \mathbf{c}'' | \mathbf{D}).$$

These results advance current knowledge of the causal or non-causal identification with fragments of information presented as conditional distributions, which is often under-studied.

### 5.2 C-Factorization-based Intermediary Criterion

We proceed to characterize an intermediary criterion where a matrix equation corresponds to a c-factorization resulting in three factors, extending the case of a pair of factors (Lemma [2]). Concretely speaking, we are interested in the matrix form of

$$P_{\mathbf{x}_\ell}(\mathbf{y}_\ell) = \sum_{\mathbf{z}, \mathbf{w}} P_{\mathbf{x}_i}(\mathbf{y}_i) P_{\mathbf{x}_j}(\mathbf{y}_j) P_{\mathbf{x}_k}(\mathbf{y}_k) = \sum_{\mathbf{z}} P_{\mathbf{x}_i}(\mathbf{y}_i) \sum_{\mathbf{w}} P_{\mathbf{x}_j}(\mathbf{y}_j) P_{\mathbf{x}_k}(\mathbf{y}_k), \qquad (4)$$

where $\mathbf{Z} \subseteq \mathbf{X}_i \cap \mathbf{Y}_j$ and $\mathbf{W} \subseteq \mathbf{X}_j \cap \mathbf{Y}_k$ are disjoint sets of variables to be marginalized (the order among the three factors is irrelevant since they are invariant up to renaming.) In addition, available distributions are of the form $P_{\mathbf{X}_j}(\mathbf{Y}_j)$, $P_{\mathbf{X}_{ij} \backslash \mathbf{Y}_{ij}}(\mathbf{Y}_{ij} \backslash \mathbf{Z})$, and $P_{\mathbf{X}_{jk} \backslash \mathbf{Y}_{jk}}(\mathbf{Y}_{jk} \backslash \mathbf{W})$.

**Theorem 3** (C-Factorization Intermediary Criterion). *Let* $\mathcal{G}$ *be a causal diagram and* $Q = P_{\mathbf{r}}$*. Let* $Q_{\mathbf{x}_\ell}(\mathbf{y}_\ell)$ *be c-factorized as Eq.* [(4)]*. Let* $\mathbf{X}_k^+$ *be a subset of* $\mathbf{X}_k$ *excluding the rest five sets,* $\{\mathbf{Y}_i, \mathbf{Y}_j, \mathbf{Y}_k, \mathbf{X}_i, \mathbf{X}_j\}$*.* $\mathbf{Y}_i^+$ *is similarly defined. If* $\mathbf{Z} \subseteq (\mathbf{X}_i \cap \mathbf{Y}_j) \backslash \mathbf{X}_k$ *and* $\mathbf{W} \subseteq \mathbf{X}_j \cap \mathbf{Y}_k$*, then* $\mathbf{Q}_{\mathbf{x}_\ell / \mathbf{X}_k^+}(\mathbf{y}_\ell / \mathbf{Y}_i^+)$*, a submatrix of* $\mathbf{Q}_{\mathbf{X}_\ell}(\mathbf{Y}_\ell)$*, becomes*

$$\mathbf{Q}_{\mathbf{x}_\ell / \mathbf{X}_k^+}(\mathbf{y}_\ell / \mathbf{Y}_i^+) = \mathbf{Q}_{(\mathbf{x}_{ij} \backslash \mathbf{y}_{ij}) / \mathbf{w}}((\mathbf{y}_{ij} \backslash \mathbf{Z}) / \mathbf{Y}_i^+) \cdot \mathbf{Q}_{\mathbf{x}_j / \mathbf{w}}(\mathbf{y}_j / \mathbf{Z})^{\dagger} \cdot \mathbf{Q}_{(\mathbf{x}_{jk} \backslash \mathbf{y}_{jk}) / \mathbf{X}_k^+}((\mathbf{y}_{jk} \backslash \mathbf{W}) / \mathbf{Z}).$$

The theorem imposes an additional constraint that $\mathbf{Z}$ should be disjoint to $\mathbf{X}_k$ compared to naively interpreting the three-matrix multiplication as two individual matrix multiplications as seen in Lemma [2] (we depict the sophisticated set relationships among $\mathbf{W}$, $\mathbf{X}$s, $\mathbf{Y}$s, and $\mathbf{Z}$ in Appendix [D] [13].) Briefly speaking, given that the summation over $\mathbf{W}$ is nested (Eq. [(4)]), $(\mathbf{X}_j \cup \mathbf{Y}_j) \backslash \mathbf{W}$ is fixed along with $\mathbf{Y}_j \cap \mathbf{X}_k$. Thus, $\mathbf{X}_i \cap \mathbf{Y}_j \cap \mathbf{X}_k$ can't be part of $\mathbf{Z}$. In other words, the constraints imposed in the original expression asymmetrically affect what $\mathbf{Z}$ can be but not what $\mathbf{W}$ can be.

The implication of this result is immediate. It was previously unknown whether the identification procedures by Lee and Bareinboim [12] (GID-PO) and Lee and Shpitser [11] (Lemma 3 and 6) taking marginal and interventional are complete. Now, we show a negative result.

**Proposition 1.** *GID-PO [12] and Lemma 3 and 6 [11] are not complete.*

*Proof.* Consider a causal graph $X \to A \to B \to Y$ and distributions $P(X, B)$, $P(A, B)$, and $P(A, Y)$. $P_x(y)$ is identified as $\mathbf{P}(y|A) \mathbf{P}(B|A)^{\dagger} \mathbf{P}(B|x)$, which is not solvable by [12, 11]. $\qquad \square$

In this section, we developed novel intermediary criteria by characterizing both chain-rule and c-factorization with respect to matrix multiplications of three matrices exploiting the pseudoinverse, which has never been employed in the context of causal identification to the best of our knowledge.

## 6 A Unifying Causal Identification Algorithm

We present a causal identification algorithm ID-ME (Alg. [1]) taking a collection of marginal, conditional, and interventional distributions $\mathbb{D}$ and causal graph $\mathcal{G}$.[6] The algorithm integrates different approaches such as generalized proxy-variable based criteria (Sec. [4] and [8, 10, 15]), intermediary criteria (Sec. [5]), and factorization approaches [23, 14, 12, 11].

Taking a causal query $P_{\mathbf{x}}(\mathbf{y})$, causal graph $\mathcal{G}$, and distributions $\mathbb{D}$, it refines the given query by removing redundant interventions based on Rule 3 of do-calculus, i.e., $P_{\mathbf{x}}(\mathbf{y}) = P_{\mathbf{x}'}(\mathbf{y})$ for a

---

[6]We omit some details on whether some variables are fixed (e.g., $P_a(B)$ versus $P_A(B)$). Without loss of generality, every distribution contains no redundant conditions and interventions, which can be obtained through repeatedly applying rules of do-calculus.

---

**Algorithm 1** ID-ME

1: **function** ID-ME($\mathbf{x}, \mathbf{y}, \mathcal{G}, \mathbb{D}$)
   **Input**: $\mathbf{x}, \mathbf{y}$ value assignments for a query $P_{\mathbf{x}}(\mathbf{y})$; $\mathcal{G}$ a causal diagram; $\mathbb{D}$ a collection of distributions
   **Output**: a formula for $P_{\mathbf{x}}(\mathbf{y})$ made with $\mathbb{D}$ or FAIL.
2: $\quad \mathbf{X}, \mathcal{G}' \leftarrow \mathrm{an}^{\mathcal{G}_{\overline{\mathbf{x}}}}(\mathbf{Y}) \cap \mathbf{X}, \mathcal{G}[\mathrm{An}^{\mathcal{G}}(\mathbf{Y})]$
3: $\quad \mathbb{D} \leftarrow \mathsf{expand}(\mathcal{G}, \mathbb{D})$ **unless** $\mathbb{D}$ is unconditional
4: $\quad \mathbb{F}^{+} \leftarrow \mathbb{F}_{\mathcal{G}'}$ **if** $\mathbb{D}$ is unconditional and nonmarginal **else** $\bigcup \{ \mathbb{F}_{\mathcal{G}' \langle \mathbf{V}' \rangle} \mid \mathbf{X} \cup \mathbf{Y} \subseteq \mathbf{V}' \subseteq \mathbf{V} \}$
5: $\quad \mathbb{F}', \mathbb{F}'' \leftarrow$ empty dictionary, $\mathbb{F}^{+}$
6: $\quad$ **for all** $P_{\mathbf{X}_i}(\mathbf{Y}_i) \in \mathbb{F}''$ and $\{ P_{\mathbf{Z}}(\mathbf{V}'|\mathbf{T}) \in \mathbb{D} \mid \mathbf{Y}_i \subseteq \mathbf{V}', \mathbf{X}_i \subseteq \mathbf{Z} \cup \mathbf{V}' \cup \mathbf{T} \}$ **do**
7: $\quad\quad$ update $\mathbb{F}', \mathbb{F}''$ with ID-RC($P_{\mathbf{X}_i}(\mathbf{Y}_i), \mathcal{G}', P_{\mathbf{Z}}(\mathbf{V}'|\mathbf{T})$)
8: $\quad$ **for all** $P_{\mathbf{X}_i}(\mathbf{Y}_i) \in \mathbb{F}''$ **do** update $\mathbb{F}', \mathbb{F}''$ with PROXY($P_{\mathbf{X}_i}(\mathbf{Y}_i), \mathcal{G}, \mathbb{D} \cup \mathbb{F}'$)
9: $\quad$ **repeat** update $\mathbb{F}', \mathbb{F}''$ with CF-INT($\mathcal{G}', \mathbb{F}', \mathbb{F}''$) and CF-INV($\mathcal{G}', \mathbb{F}', \mathbb{F}''$) **until** $\mathbb{F}'$ not changed
10: $\quad$ **return** exact-cover($\mathcal{G}', P_{\mathbf{x}}(\mathbf{y}), \mathbb{F}'$)

---

minimal subset $\mathbf{X}' \subseteq \mathbf{X}$. Further, $\mathcal{G}'$ a copy of causal graph is obtained through pruning $\mathcal{G}$ only to the ancestors of $\mathbf{Y}$. However, note that the original $\mathcal{G}$ is kept intact to be used in the proxy criteria later where proxies can be non-ancestors of $\mathbf{Y}$, e.g., Fig. 1b. Then, the algorithm proceeds the following three parts: 1) expanding the given distributions $\mathbb{D}$; 2) c-factorizing the query $P_{\mathbf{x}}(\mathbf{y})$ and identifying each factor; and 3) combining identified c-factors $\mathbb{F}'$ to elicit the causal query.

First, the algorithm *expands* the given distributions (expand in Line 3). A given conditional distribution $P_{\mathbf{Z}}(\mathbf{V}'|\mathbf{T})$ can be understood as a distribution $P_{\mathbf{Z}}(\mathbf{V}')$ under a selection bias, which might hinder identification of a c-factor. By multiplying another distribution equal to, e.g., $P_{\mathbf{Z}}(\mathbf{T}''|\mathbf{T}')$, we can obtain a distribution with more outcome and less bias, $P_{\mathbf{Z}}(\mathbf{V}', \mathbf{T}''|\mathbf{T}')$. Chain rule closure [11] describes a state of a set of distributions where no new distribution with a smaller condition can be added to the collection. Function expand contains a procedure for chain rule closure along with chain-rule matrix inversion (Lemma 1) and chain-rule intermediary criterion (Lemma 4), enjoying the matrical approach especially relevant to conditional distributions.

The next part enumerates c-factors (Line 4) and attempts to identify every c-factor based on available distributions (Lines 5–8). An algorithm for identifying a c-factor given a *single* unconditional and non-marginal distribution has been thoroughly studied (e.g., Identify [29] and ID [23]), and is a building block for general identification [2, 14]. Dealing with a marginal or conditional distribution, one should consider (i) whether the condition in the conditional distribution in $\mathbb{D}$ can be negligible with respect to identifying the c-factor or (ii) whether matrix equations can be utilized (e.g., simple inverse or generalized proxy criteria). ID-RC (Line 7) is an identification module modified to handle a conditional distribution [4, 11], and PROXY (Line 8) refers to generalized proxy criteria.

The last part (Lines 9–10) attempts to map the identified c-factors to the causal effect $P_{\mathbf{x}}(\mathbf{y})$. Once a subset of c-factors are identified from the previous step, the algorithm proceeds to examine whether some of the unidentified factors in $\mathbb{F}''$ can be further inferred from the individually identified factors $\mathbb{F}'$, i.e., whether a simple inverse (CF-INV implementing Lemma 2) or the intermediary criterion over c-factors (CF-INT implementing Thm. 3) can be invoked (Line 9). Then, we finally elicit a causal effect if a valid combination of identified c-factors exists (exact-cover in Line 10 checking Eq. (2)).

The algorithm coherently integrates existing methods and the newly developed machinery in the previous sections. Whenever ID-ME returns a formula, evaluation of the formula leads to the quantity $P_{\mathbf{x}}(\mathbf{y})$. Further, ID-ME *strictly* subsumes the union of aforementioned methods in a way that not only returns it a formula whenever some of the methods returns one given the same problem instance but also it can output novel formulae for the problems that cannot be answered by any of them as demonstrated through examples and by Prop. 1.

**Theorem 4.** *ID-ME is sound.*

**Theorem 5.** *ID-ME strictly subsumes proxy criteria [8, 15], GID(-PO) [14, 12], mID, or eID [11].*

Finally, we discuss the time complexity of ID-ME through examining the complexity of algorithms concerning a subset of problem instances ID-ME can handle. First, the state of the art identification conditions with distributions marginal, experimental, but unconditional [12, 11] involve finding a latent projection where the corresponding set of c-factors are all identified. It is conjectured that it requires time exponential in $|\mathbf{V}|$ [12]. Next, dealing with conditional distributions through expanding

them takes time exponential in $|\mathbf{V}|$ (e.g., chain rule closure [11]). As a result the algorithm generally runs in time exponential in $|\mathbf{V}|$, which we speculate that this complexity reflects a fundamental trade off between expressive power and tractability within causal inference, as already observed in [12, 11].

## 7 Conclusion

In this paper, we studied the use of matrix equations in causal identification given general distributions. In particular, we characterized matrix equations made of distributions driven by graphical constraints, deepening our understanding on algebraic constraints imposed by the graph (Lemmas 1 and 2 in Sec. 3). We generalized proxy-based identification methods to cover a broad spectrum of problem instances beyond the identification problems with an observation study and optional external study (Thms 1 and 2 in Sec. 4). We developed novel intermediary criteria that identify a query by utilizing the pseudoinverse of the center matrix within the multiplication of three matrices (Lemma 4 and Thm. 3 in Sec. 5). Finally, we provide a causal identification algorithm (Alg. 1 in Sec. 6) integrating several existing identification results to produce an identification formula for a causal query given a causal graph and distributions that can be marginal, experimental, and conditional harnessing the power of matrix equations and (pseudo)inverses. As a future research direction, it will be crucial to devise a statistically efficient estimator for identification formulae involving matrix inversion.

## Acknowledgments and Disclosure of Funding

Sanghack Lee was supported by the New Faculty Startup Fund from Seoul National University. Elias Bareinboim was supported in part by funding from the NSF, Amazon, JP Morgan, and The Alfred P. Sloan Foundation.

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
