# OpenReview forum: "Causal Identification with Matrix Equations"
_NeurIPS.cc/2021/Conference — NeurIPS 2021 Oral_

### Official Review · Reviewer_h23K · 2021-07-15

**Rating:** 6
**Confidence:** 3

**Summary:**

This paper considers the causal effect identification problem, and proposes an approach making use of both graphical criteria and matrix equations. More precisely, the authors connect the graphical and matrical approaches by characterizing matrix equations of probability distributions driven by graphical constraints in a causal graph. Building on this new characterization, the authors generalize proxy-based criteria and devise novel intermediary pseudoinverse criteria so as to identify a causal effect by utilizing the general inverse of a matrix and diverse collection of distributions.

**Limitations And Societal Impact:**

Please see the main review.

**Main Review:**

This paper builds up on the work of [Miao et al.], and generalizes their results to some extent. They specially draw attention to the following points: 1) in certain circumstances, the invertibility condition of the matrices in causal identification can be dropped given certain d-separation relations, and we can work with pseudo-inverse rather than inverse of these matrices. 2) Matrix factorization of the desired causal effect can be extended to more than two factors (as in [Miao et al.]). 3) the causal identification problem when partial data or conditional distributions are provided is a line of research that needs more attention.

The authors relax some of the assumptions, namely invertibility of certain conditional probability matrices and also some d-separation relations, and show that the causal effect is still identifiable under certain conditions.
They provide a generalization of the “MGT”. However, the extra conditions they need in theorem 2, namely the surrogate set, converts the problem to the original one.

It would be better to include the definition of an “admissible set”, or at least refer to the works by Pearl or Greenland et al.

They are assuming throughout the paper that the data is from an interventional distribution P_r. I understand that this is to show that the results are general and the data doesn’t need to be observational, but this could be pointed out as a remark rather than including the irrelevant notation r, P_r, etc. in every result which reduces the readability.

The idea of the paper and the big picture sounds interesting, but the contributions on the theory side are not major in my opinion. The results are mainly based on a few examples and only one-way theorems that indeed generalize MGT in a way, but do not contain solid improvements/contribution.


 The big picture and the motivation are interesting, but the contribution and the generalization scheme are not profound enough. It sounds like a rephrasing of the main problem in a more general case but with conditions that converts it to the main case with less effort.

**Time Spent Reviewing:**

3

---

> ### Author Response · Authors · 2021-08-10
> **Response to Reviewer h23K**
>
> Thanks for reviewing our paper and finding our big picture and motivation interesting. We would like to address your critical assessment that the paper lacks solid improvements and our profound contributions.
>
> We strongly believe that our work presents technically profound results and solid contributions which we elaborate next. There are two families of identification methods known in the literature, c-component-based [2, 4, 9, 10, 11, 18, 22] and proxy-based [6, 8, 12]. The former is quite general and strong in terms of guarantees, noting that one can provide any graph and any distribution, and the algorithm will decide whether an effect is identifiable in a non-parametric fashion. Still, this assumes that the measurements across datasets are consistent, i.e., over the same set of endogenous variables. The latter (proxy-based) is quite powerful and says that, with mild assumptions regarding the relationship between the unobserved variables and its proxy, some new effects can be identified. Still, these results are circumscribed to some specific causal graphs and some special set of assumptions. Both approaches have been studied mostly in isolation, but they are complementary, which is the departing point of the investigation proposed in this paper. We develop the first general method combining both approaches from the first principles. There is no benchmark method that could compare against the proposed method since it handles cases that individually these approaches cannot handle. Also, remarkably, these results (Sections 3 and 5) are highly non-trivial since “combining” these families is not a matter of handling one in tandem with the other, but it requires a serious understanding of the foundations and how these methods work so that they can be integrated in a principled and systematic way (e.g., Theorem 3). In fact, an entirely new formal framework has been developed to allow this connection to be realized (Section 6). After all, we hope that the feeling that this is a simple generalization of Miao et al. [12] can be overcome. It’s a generalization of perhaps Miao, Kuroki, Pearl, and at least another 10 papers (some of them cited above) that came before and solved special instances of the identification problem, in a fragmented way, and now we have a unifying treatment. After all, Algorithm 1 (ID-ME)  is the most general identification algorithm known to date.
>
> The use of $P_{\mathbf{r}}$ throughout the paper is to ensure that the presented results are not only applicable relative to $P$ but also generalize to other experimental distributions. Although adding a remark can certainly simplify the text, we are afraid that replacing $P_{\mathbf{r}}$ to $P$ may misrepresent our results and mislead some of the readers since $\mathcal{M}$, $\mathcal{G}$, and $P$ have well-established semantics in the literature. If we present, e.g., Lemma 4 (Chain-Rule Intermediary Criterion) using $P$ without $P_{\mathbf{r}}$, some of the readers might think that the result is only applicable to $P$, which isn’t true.
>
> In terms of the “minor comments”, we will define the 'admissible set' formally.

---

### Official Review · Reviewer_6Pv4 · 2021-07-17

**Rating:** 6
**Confidence:** 3

**Summary:**

Two primary methods exist for graph based identification. The first relies only on the assumptions in the graph while the second exploits assumptions  between unobserved confounders and the observable variables via proxy variables. This paper develops a new approach by combining both these methods. The new approach thus developed is able to solve problems that are not solvable by any of the methods alone.

**Main Review:**

“We may employ conditional to emphasize…” Line 80, page 3. This sentence is not clearly worded.

It seems like P_{a,B}(C,d) should be a |X_C| x |X_B| matrix instead of the |X_B| x |X_C| as specified on line 114, page 3.

“A special case of the lemma is appeared in” Line 125, page 4. Include a reference to double proxy setting on page 6 where MGT is described.

Section 3: Your audience will already be familiar with the usual (non-matrix) chain rule, adjustment criterion etc. Why not replace this whole section with a table with the usual definitions in the first column, matrix definition in the second and a toy example for the  matrix definition in the third column?

Lines 146-161: Please try and reword this part. “As a result, we can obtain...” I am having difficulty understanding the last two sentences. An example with matrices would have been very helpful.

A careful checking of proofs and claims in this paper will require an effort spanning multiple days. Unfortunately, this wasn’t possible. This work seems to be apt for a journal where it would benefit by multiple rounds of review by the same set of people. Additionally, there will be no space constraint in a journal. So the authors will be able to include examples and results, all in one place thus making the paper easy to follow.

The referencing style is not uniform throughout the paper. In the abstract, it is names such as Lee & Barinboim, 2020, Kuroki & Pearl, 2014 and Miao et al 2018. In the paper the authors revert to numbers. Other relevant paper that precede the work by Kuroki & Pearl include,

Greenland, S.andLash, T.(2008). Bias analysis.In Modern Epidemiology (K. Rothman, S. Green-land and T. Lash, eds.), 3rd ed. Lippincott Williamsand Wilkins, Philadelphia, PA, 345–380

Carroll, R.,Ruppert, D.,Stefanski, L.andCrainiceanu, C.(2006).Measurement Error in Nonlinear Models: A Modern Perspective. 2nd ed.Chapman & Hall/CRC, Boca Raton, FL.


**Time Spent Reviewing:**

---

> ### Author Response · Authors · 2021-08-10
> **Response to Reviewer 6Pv4**
>
> Thanks for constructive feedback to improve the paper. Once we reflect your suggestions into the paper, the paper will be much more readable and suitable for conference publication. Let us address the points you raised one by one.
>
> (Sentence in Line 80): We meant that we might say a distribution is conditional (or a conditional distribution) to highlight that $\mathbf{W}$ can be non-empty (but not excluding the empty case, i.e., for marginal distributions). Hence, conditional distributions subsume the treatment for marginal ones. We note that the latter (marginals) are the most common case investigated in the literature. Our goal is to allow the input distributions to be “heterogeneous”, in the sense of subsets of the observables, say $P(\mathbf{V}_1)$, $P(\mathbf{V}_2)$, $P(\mathbf{V}_3 | do(\mathbf{Z}))$, etc, where $\mathbf{V}_1,\mathbf{V}_2,\mathbf{V}_3 \subset \mathbf{V}$.
>
> (Matrix dimension): You are right, thanks for catching the typo.
>
> (Reference for Line 125): We will add the reference.
>
> (Section 3 as a table): We will see how to improve Section 3. We feel somewhat uneasy about replacing the whole section with a table without hurting legibility, but would be curious to hear the opinion of the other reviewers. We note that this section contains three unnumbered subsections with important pieces of background. Specifically, the first section is not just about a chain rule but with marginalization and conditional independence relationships involved. Further, presenting the adjustment criterion is important given their use in Section 4. Finally, c-factorization needs careful explanations to grasp the gist of Lemma 2. Again, we think the presentation can be improved and will try to do so if the paper is accepted and an additional page is allowed.
>
> (Lines 146-161 before Lemma 2): Thanks for the suggestion. We are considering adding a bit more explanations together with a figure of matrices as we commented for Reviewer rBKj.
>
> (Referencing Style): The inconsistency in the abstract is intentional. We considered numbered references for abstract is not desirable since people will often read abstracts without an access to the main text.
>
> (Additional References): Thanks for recommended references. We will include Carroll, et al. Although we have cited Greenland and Lash (e.g., Line 41), we will ensure that the paper is cited wherever appropriate (e.g., in Section 4).

---

### Official Review · Reviewer_yA7o · 2021-07-19

**Rating:** 6
**Confidence:** 3

**Summary:**

This paper considers causal inference in the discrete case, where probability axioms can be represented with matrix equations. Specifically, the paper characterizes the relationships between certain graphically driven formulae and matrix multiplications. With such characterizations, the authors then broaden the spectrum of proxy variable-based identification conditions and further propose novel intermediary criteria based on the pseudoinverse of a matrix.


**Limitations And Societal Impact:**

yes

**Main Review:**

This paper studies a very important problem: causal inference with matrix equations in the discrete case. Such matrix equation characterization has been used in previous work, e.g. in [12]. So, with respect to the matrix equation, it is not a novel characterization. Instead, the novelty of this paper is that it broadens the spectrum of proxy variable-based identification conditions and proposes intermediary criteria based on the pseudoinverse of a matrix.

Some parts of the paper are not very easy to follow and I did not check the proofs.  A question: whether such generalized proxy-based conditions can be extended to the continuous case, like that in [12]?

There is no experimental validation in the paper. To show the effectiveness of the proposed algorithm, I suggest that the authors add some experiments.

Updates:
I have carefully read the authors' rebuttal. I understand it is not so intuitive to solve my concerns, but I still think the authors should take these concerns more seriously.


**Time Spent Reviewing:**

2

---

> ### Author Response · Authors · 2021-08-10
> **Response to Reviewer yA7o**
>
> Thank you for reviewing our paper. The readability of the paper will be improved with more explanations using an additional page, if the paper is accepted.
>
> ### Extension to Continuous Case
> Although we strongly believe that the 'completeness condition' presented in [12] (different from the completeness of an identification algorithm) can be generalized to the matrices we inverted in the paper, it is not as obvious as it may seem when dealing with sets of variables and generic causal graphs. For instance, the condition in [12] involves not only $W,Z,X$ for the invertibility of $P(W|Z,x)$ but also $Y$ and $U$ in the graph should be accounted for. Investigating this line of research would be an important next step.
>
> ### Experimental Validation
> Empirical causal effect estimation based on our algorithm would be valuable to assess the practical aspect of our identification algorithm.
> We believe that developing an efficient and robust estimator involving matrix (pseudo)inverse will be extremely challenging. Hence, it is hard for us to address both identification and estimation in a single paper given the complexity of the identification conditions already, as discussed in the paper.  Still, we do agree that this is a quite significant research question.

---

### Official Review · Reviewer_rBKj · 2021-07-19

**Rating:** 6
**Confidence:** 2

**Summary:**

This paper presents a synthesis between literatures on using matrix equations to characterize causal effect identifiability, and using proxies/surrogates to obtain identifiability when the effect of interest is not identifiable using the canonical ID algorithm. The authors highlight examples of identifiability under the various identification paradigms. They then describe notation used for their synthesis and highlight how it applies to each of the known cases. The main results entail an algorithm for determining identification of a broad class of effects via the matrix equations paradigm. The authors show that some key existing results, which had been proven sound, are not complete and show that their approach, while not yet proven complete, generalizes the existing approaches for the literatures studied.

**Limitations And Societal Impact:**

Not really, but this paper is so theoretical that I don't really think it matters.

**Main Review:**

As is the case with all identification papers, this one is dense and a lot of the notation is hard to parse/get lost in. Unfortunately I don't have any advice for the authors on how to simplify the notation. It is quite clear the authors have already thought about this very carefully (section 2 stuck out to me as a particularly efficient characterization of the necessary notation). I found that the paper flowed nicely and the building of one idea off the other was generally reasonably easy to follow (or, rather, as easy as it could be for this sort of paper). I also appreciated the authors' attempts to add clarity through examples and illustration, such as with the Euler diagram in Fig. 2.

Candidly, this paper relies pretty heavily on 4 papers (citations 9-12) I am only familiar with at a high level and so I do not know that my review will serve as the best assessment of this paper's quality.

Detailed comments for the authors to consider follow:

**Presumptions about audience knowledge** -- given that this paper is under consideration at NeurIPS (a wide and not necessarily deep causal audience), I suspect many ideas from background won't be well-known or well-understood. Despite the authors' clear attempts at clarity, there are some opportunities for further improvement. E.g. In line 56/57, the authors state that Px(y) is not identifiable in Fig. 1d by either the proxy approach nor the matrix equations approach. Why is this the case? What is the hedge/district/(insert poorly named, non-intuitive graphical construct from the literature) that "witnesses" non-identifiability here?

This also applies to the statement in line 194. What is it about the structure of the referenced graphs that makes it infeasible to use the approach described in lines 186-193?

**Explanations of Math** -- Similar to the above point, there are cases where it would help to explain a little bit more clearly what is going on from step t to step t+1 in terms of classical notation. For instance, for the chain rule below line 119, I had to scribble on my copy of the paper for a few minutes to say "ok...so they split out b like that...then...??...they can sum it out...then..." to be able to verify my understanding (on the other hand, once I had that understanding, I was basically able to prove Lemma 1 myself so perhaps it's your intention that there be more of a learning curve?). My opinion is that when you're introducing a new (or not-widely-used) notation, it's important to make a bit more clear what's going on by not skipping as many steps.

**Invertibility Assumptions** -- Throughout it was somewhat hard for me to grasp the interpretation of (non-)invertibility. Is it the case that non-invertibility --> non-identifiability? I don't think so because the results here are not complete. Some intuition here would be helpful, especially in the exposition-y sections (mid-section 3 is where I've written this question in the margin).

**Line 165** -- the expression below line 165 is hard to read, mostly because of all the alternating between \ and /. Is the / meant to represent some sort of division (multiplication by inverse?)? The proof of Lemma 2 in the appendix didn't shed any light on this.

**Section 6** -- I actually very much appreciated the line-by-line description provided throughout section 6. This is missing from most ID-related papers.

**Time Spent Reviewing:**

7

---

> ### Author Response · Authors · 2021-08-10
> **Response to Reviewer rBKj**
>
> Thanks for reviewing our paper and acknowledging our presentation effort through the flow, examples, figures, and other elements raised. This is quite a non-trivial task since while introducing the basics in a gentle way is certainly expected, we also need to allocate a reasonable amount of space to present the novel results of the paper. Obviously, this is not a survey paper, and some of the top and most technical results in the field of causal inference have appeared in NeurIPS for the last decade or so. After all, we hope not to be penalized for doing technical work and building on the state of the art causal inference literature. Again, thank you for your patience and we will try to address the main issues raised in your review in the sequel.
>
> ### A Negative Example in Fig 1d:
> The proxy criteria are applicable to certain types of graphs satisfying a set of assumptions (looks similar to Fig 1b and c), which is not the case for Fig 1d. In general, identification algorithms treat unobserved variables as bidirected edges (i.e., the latent projection of the graph). In this case, the resulting projection of Fig. 1d will contain a hedge (i.e., $U_1$ is marginalized out as a bidirected edge between $X_1$ and $Y$). Hence, the effect is not identified. The phrase "not identifiable" might be confusing as it describes the quality of the query not the result of algorithms. A better choice would be "not identified." We will carefully add explanations for Fig 1d.
>
> ### Negative Examples in Fig 3a,b in Line 194:
> $(X \perp W \mid U)$  is one of the central assumptions for the single proxy criterion by Greenland and Lash [6]. With $X$ and $W$ directly connected in the examples, we cannot make use of an equality $P(W|U,x)=P(W|U)$ where $P(W|U)$ is available. Part of Eq. 3 to compute $P(y,U|x)$ and $P(U|x)$ requires the equality, and hence, the criterion expressed in Eq. 3  is no longer applicable to the examples in Fig 3a,b.
>
> ### Explanations of Math
> Thanks for pointing this out and we do not . With more space available once the paper is accepted, we will be able to provide clearer explanations for newly introduced symbols.
>
> ### Non-invertibility and Non-identifiability:
> Consider an expression $A=B^{-1}\cdot C$ where $A,B,C$ are matrices. Non-invertibility of $B$ implies that the system is underdetermined, and $A$ is not identified given $B$ and $C$. However, there is no formal statement of non-identifiability in Greenland and Lash [6], Kuroki and Pearl [8], or Miao et al. [12], which are the references we used as a baseline to this work. Although we similarly do not make the claim that the non-invertibility implies non-identifiability (results represented in Sections 3 and 4), we conjecture that they are indeed necessary, and therefore complete.
>
> ###  Parsing Lemma 2:
> The backslash $\setminus$ is a  conventional set subtraction operation. We will make this explicit in Section 2 (Preliminaries). It may be applied to a set of values as well (Line 79.) A forward slash $/$ is the one introduced in Line 161. For example, Let $\mathbf{a}$ be $\\{ a_1, a_2 \\}$ values of two variables $A_1$ and $A_2$. With $\mathbf{B} = \\{ A_2, A_3 \\}$, $\mathbf{a}/\mathbf{B}$ yields a pair of value $a_1$ and variable $A_2$. To help readers understand Lemma 2, we would be able to visualize the three matrices involved in Lemma 2 more clearly so as to explain better what variables should be fixed and why.
>
> ### Line by Line explanation in Section 6:
> Thanks for the positive comment. In the camera-ready version, we will even be able to add a running example in the main text as well.
>
> We will be able to address issues you raised and the paper will be more accessible to readers without losing the technicality.

---

### Decision · Program_Chairs · 2021-09-28

**Decision:**

Accept (Oral)

**Comment:**

This paper aims to unify (everywhere) identification theory based on graphical models, and (generic) identification theory based on proxies.

The paper was very difficult to follow, but after a fair bit of discussion, reviewers saw merit in the results presented, despite the difficulty with presentation.

**Consistency Experiment:**

NeurIPS has a long history of experimentation. In 2014, NeurIPS ran an experiment in which 10% of submissions were reviewed by two independent committees to quantify the randomness in the review process. This year, we repeated a variant of this experiment to see how the quality of the review process has changed over time.  This paper was part of the experiment and was therefore assigned to two committees (consisting of reviewers, an Area Chair, and a Senior Area Chair) that reached independent decisions.  If both committees made the same recommendation, this recommendation was followed. If a single committee recommended acceptance, the paper was accepted (with the exception of a few cases in which the other committee identified what we considered a fatal flaw, e.g., an error in a key result).

This copy’s committee reached the following decision: **Accept (Poster)**

The other committee assigned to the paper recommended **Accept (Oral)**.  You can find the other set of reviews, along with any follow up discussion with the authors here:
https://openreview.net/forum?id=wfJCeMS-jH